# Transformation for a Post-Pandemic World: Exploring Social Innovations in Six Domains

Manuel Laranja [1] and Hugo Pinto [2,3,*]

1   ISEG—Lisbon School of Economics and Management, University of Lisbon, 1649-004 Lisbon, Portugal; mlaranja@iseg.ulisboa.pt
2   Centre for Social Studies, University of Coimbra, 3004-531 Coimbra, Portugal
3   Faculty of Economics, University of Algarve, 8005-139 Faro, Portugal
*   Correspondence: hpinto@ces.uc.pt

**Abstract:** The world is suffering from a myriad of challenges. These are not only a direct result of the COVID-19 pandemic but also the evidence of a series of structural problems that existing socio-economic structures have. Inspired by real examples of social innovations and based on a selective literature review, this article debates six domains of opportunities for social innovation. These domains refer to well-being, finance and banking, work, technology, learning and education, and leadership and governance. The article ends with crucial implications for implementing change and a sustainable transition.

**Keywords:** social innovation; transformation; transitions; finance; technology; governance; pandemic; well-being

## 1. Introduction

The crisis triggered by COVID-19 has exposed many weaknesses of the current socioeconomic system: job insecurity for many people, the increase in remote work, the emphasis on essential professions needed to ensure public and vital services, the difficulties felt by healthcare systems in responding to the pandemic due to years of budget cuts and austerity measures, among other issues. Millions of children die every year from preventable causes and more than 800 million people are undernourished. At the same time, biodiversity and ecosystems are constantly being degraded and greenhouse gases continue to soar, leading to anthropogenic climate change and causing sea level rise, stronger storms, droughts and wildfires that devour entire regions. While the world economy produces more than ever before, it fails to take care of humans and the planet. In addition, we need to be aware of the exploitation of the crisis by anti-systemic movements, which welcome measures such as the rise of mass surveillance and invasive technologies, border closures, and restrictions on the right of assembly.

The current social, economic and environmental changes, the ones stemming from the COVID-19 pandemic and the ones that were presented before, bring new challenges and demand reflexive analysis. To a certain extent, the pre-pandemic global surge of fundamentalism, xenophobia and authoritarianism (e.g., in the EUA, Brazil, Hungary, Poland) underlines the same phenomena: the struggle as a society to respond to the need for deep social changes, i.e., the lack of capacity to deal with entire social system changes over longer processes. Instead, we continue to see change as a "fast forward" of the past.

Hence, there is a need for a new vision of social innovation. The field of social innovation has emerged over the last two decades, attracting interdisciplinary research and interest from policymakers, third-sector organizations, and businesses [1]. Social innovation refers to an idea that deliberately attempts to better satisfy explicit or latent social needs and problems, resulting in new or improved capabilities, and in the transformation of social

and power relations, aiming at social change and the establishment of new social practices that positively affect the lives of individuals [2].

The need to respond to the current social changes has triggered a growing international movement of people and organizations at the intersection of the ecology, civil rights and participatory democracy social movements. These relatively new initiatives are often test beds for new forms of cooperation and solidarity that appear to be flourishing. For example, there is now wider appreciation of basic societal services like universal income and health. There is also progress in the adoption of new online work models, such as online education. The pandemic has also led to unprecedented government actions, demonstrating what it is possible to achieve when there is a will to act. For example, the reshuffling of public budgets, the expansion of social security systems to cover for temporary unemployment, the rapid public health campaigns and investments, particularly in the testing of those infected with the SARS-CoV-2, and the outstanding support awarded to R&D in the search for a possible vaccine.

The pandemic crisis brought about a new opportunity to start a transition towards a radically different kind of society. However, if we are not fated to go back to normal, we need disruptive changes and a new vision for social innovation. The framework proposed in this article assumes six key dimensions to reflect around needs and opportunities for profound changes and social and economic innovations (Figure 1). These dimensions are also directions of research for our non-systematic literature review, which is meant to be explorative and informative rather than all-encompassing, and directed to specific research questions.

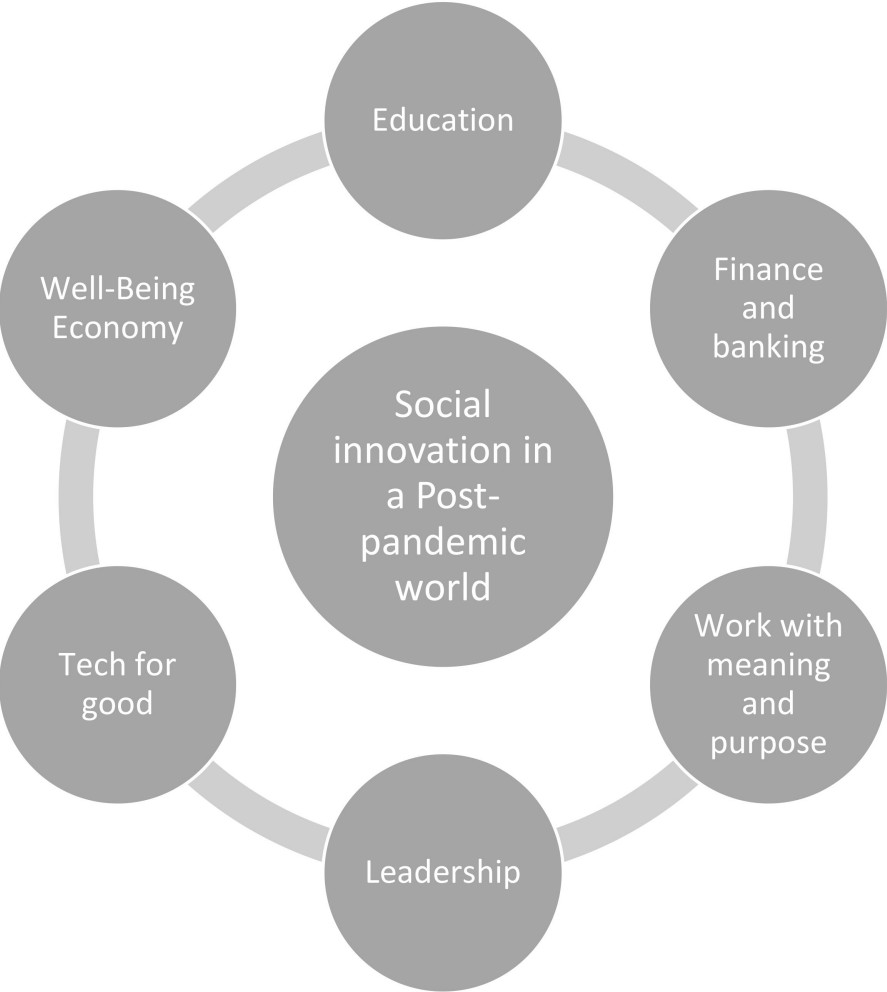

**Figure 1.** Key dimensions for meaningful change. Source: Own elaboration.

In each key dimension, social innovation may be a partial answer to tackle new challenges. In addition, existing tentative responses that we identify in each dimension may be interesting points for reflecting on how to trigger wider socioeconomic transformations.

The article is organized as follows.

Section 2 will be dedicated to reflections on the major social innovations needed to make way for what is called the well-being economy. In our view, the well-being economy corresponds to a post-growth approach to economics based on the assumption that we cannot continue on a path of endless economic expansion that relies on externalizing the cost of natural capital onto the planet and future generations. Major changes are needed in order to slow down, while sustaining good quality of life for all (and not solely for the more developed countries). Following from the previous section, Section 3 will look at finance and banking as vehicles for social innovation. The current finance system, despite accumulating an oversupply of money and capital, focuses its funding in areas that produce high financial returns, but low environmental and social returns. However, there are cases of banks and funds that choose to do otherwise. We need to reflect on these examples, and how they can inspire a larger transformation of the financial sector. In Section 4, we will reflect on how social innovation may help set new ways of working, more specifically, working with meaning and purpose. Opportunities for social innovation related to work cannot just focus on means to find jobs for the jobless, but should also aim to rethink the notion of work by factoring in meaning and purpose. Another issue that needs to be analyzed is how technology can serve the needs of social innovation. While we often see science and technology presented as capable of solving anything, digital technologies are creating new problems that need to be addressed. In Section 5, we will critically look at the uses and misuses of technology, in particular digitalization and internet technologies. Further to the use of capital and technology in support of social innovation, in Section 6 we will also reflect on education. Education is usually considered as the main "social elevator", i.e., the main mechanism for upward social mobility. However, the education system itself is now being questioned more than ever before. There is a poor fit between what the education system teaches and the increasing complexity and interdependency of social and economic phenomena. With the use of technology, the learning process is also changing to a much more autonomous, self-determined process, which poses new challenges for the education system. In Section 7, we will reflect on the role of social innovation in promoting changes in leadership and new governance mechanisms. Failure of current leaders to anticipate change suggests that the established forms of leadership and governance are no longer working. There is a new need for leaders capable of advancing transformation processes that prioritize human, social and environmental issues. Finally, the document presents some concluding remarks regarding opportunities for social innovation along the key dimensions considered.

## 2. The Well-Being Economy

Many of the social and economic problems we witness today have their roots in a deep ecological divide between humans and nature. While it is true that there have been significant improvements in eco-efficiency that will continue in the context of the European Green Deal, these gains will not compensate for the current rate of economic expansion, which will lead to higher natural resources usage.

Today, we use more ecological resources than nature can regenerate, and carbon dioxide emissions into the atmosphere continue to grow. At the time of writing, humanity required the equivalent of 1.7 Earths to provide the resources needed to fuel the economy and absorb all the waste, meaning that it takes the Earth one year and seven months to regenerate what we use in a year [3].

Using 12 social-economic indicators and 12 Earth-system indicators, Will Steffen and others [4] argue that since the 1950s, the dominant feature of the global socioeconomic system is that the economic activity of the human enterprise is growing at a rapid rate. A "great acceleration" in human activity is causing profound changes in the structure and

functioning of the planet's physical, chemical and biological systems, as clearly illustrated by Earth-system indicators. Human actions have been a primary cause of the climate change observed in the past decades, affecting vital ecosystems such as agriculture, water management, transportation, fishing, biodiversity and biological conservation. For example, difficulty in accessing freshwater is likely to increase six-fold in the future, and one-third of agricultural land has disappeared over the last decades.

The mismatch between infinite economic growth and finite (natural) resources is therefore the consequence of a deep disconnect between man and nature. Accordingly, in all modern economic theory, "nature" is taken as a commodity. However, as claimed by Karl Polanyi [5], nature is not produced by humans and, therefore, should not be treated as a "commodity fiction".

On the other hand, there appears to be a clear association between the value we attribute to social and environmental dimensions and easy access to green community gardens [6]. Our disconnect with nature can also be seen in the so-called "green innovation proposals". According to this approach, aspects of technological innovation such as renewable energy or carbon sequestration can provide solutions that enable people to keep or even enhance their current lifestyles while contributing to the creation of new jobs and maintaining economic growth. Jeremy Rifkin, author of "The Third Industrial Revolution" [7], is particularly confident about this perspective, arguing that open-access online tools, individual means of production (new spaces for services and production), and technologies for renewable energy will eventually converge to a more efficient and democratically distributed use of natural resources and energy, similar to what has happened with access to the internet. However, the issue is not how we distribute the usage of natural resources but how we overuse natural resources. In addition, the beneficial effects of green innovation may not be available to all, and instead be the privilege of a few in the Western countries.

With the same goal in mind, Ernst Ulrich von Weizsäcker [8] argues that technology can help achieve "factor five", i.e., around 80% improvement in resource and energy productivity and, therefore, contribute to retooling the economic system, massively boosting wealth for billions of people around the world, and helping solve climate change crises. Much in the same way, the Circular Economy, or Cradle to Cradle as it was called in 1992, proposes to replace the current Linear Economic (take, make and throw away) paradigm with closed loop cycles in materials and energy, greatly contributing to recycling and waste reduction.

However, both the green and the circular economy approaches, while reducing waste and improving natural resources productivity, still assume that the current social economic system will continue to grow, either by using technology fixes and/or by re-organizing materials and energy in circular chains. Nonetheless, the gains these approaches bring will not sustain a much-needed stable relation between Earth's natural ecosystems and human socioeconomic development. These approaches and their variants do not account for the true value of natural resources. According to a report by the environmental consultancy company Trucost [9] on behalf of the Economics of Ecosystems and Biodiversity (TEEB) initiative and sponsored by the United Nations Environmental Program—UNEP, no industry would be profitable if the true costs of "natural capital" were to be properly accounted. For example, if we accounted the costs of green gas emissions, of cleaning contaminated soil and water, and of funding the added health services in support of the spread of diseases related to industrial products, no industry would make any profit. Research by the World Economic Forum [10] suggests that USD 44 trillion of economic value generation—over half the world's total GDP—is moderately or highly dependent on natural resources.

Nature loss matters for most businesses through impacts on operations, supply chains and markets. In other words, the current economic paradigm relies on the unassessed value that nature creates, and that value is most likely higher than the economic value of all products and services produced. Nevertheless, the costs and risks of natural capital are still not considered. Traditional economic theory assumes that value is created inside businesses

(the "production function") and ignores natural capital as well as the value created by other public goods such as education and social welfare development. Creation of value involves activities of the community and of the people as citizens, and not just business activities serving the purpose of generating income, which could be best described as commerce. Current economics needs to come back to the original notion of "common house" i.e., the economic system as a whole, including its social, cultural and ecological impact.

Perhaps also because of this deep disconnect with nature (and with other "commons"), the discussion about the COVID-19 pandemic and the opportunity for other diseases to spread among humans usually misses the links between health, the loss of biodiversity and social-economic development. The World Health Organization (WHO) has already estimated that, globally, 4.2 million people die each year from outdoor air pollution, and that the impacts of climate change are expected to cause 250,000 additional deaths per year between 2030 and 2050. Experts warn that with further severe degradation of ecosystem functions (a scenario that is to be expected under the current economic model), chances of future and even stronger virus outbreaks are realistic.

Another aspect that appears to be at the root of main social and economic problems is that the current economic system has not been able to reduce inequality. While the richest 1% of people in the world (adults with an income over USD 500k) own 40% of the world's wealth, 50% of the world's population own just 1% of the world's household wealth, resulting in wealth concentration in one part of the society and unmet basic needs in another. The gap between the highest income earners and the rest continues to grow, while upward social mobility has declined. With the global financial crisis of 2008, people living with less than USD 2.50 a day rose to 2.5 billion and people living in extreme poverty, meaning that they live with less than USD 1.25 a day, increased to 1.3 billion.

The 2018 report on "Living conditions in Europe" published by the Eurostat [11] highlights that:

- 21.7% of the EU population—or some 109 million people—was at risk of poverty or social exclusion.
- Almost half (45.8%) of the EU population lived in single person households with dependent children, and was at risk of poverty or social exclusion.

Overall, the deep ecological and social imbalances briefly referred above challenge the current economic paradigm—often called "growth" paradigm—and highlight the need to rethink the economy and reflect on current general eco-modernist ideas. Relying on technological innovation and global markets to solve humanity's challenges may be a dangerous illusion. For example, meeting global objectives such as those of the 2015 Paris Agreement will only be possible if early industrialized countries (Western countries in particular) include a de-prioritization of economic growth in policymaking as a precondition for re-embedding their economies and societies into the planetary natural resources renewal limits. This will require nothing less than a drastic slowdown in lifestyles. We need a "post-growth" approach based on the assumption that the planet will, under no circumstances, be able to sustain prosperity as we know it today in developed countries, and as it is pursued by emerging economies. Proponents of a post-growth approach [12] claim that we cannot continue on infinite economic expansion that relies on externalizing the costs of natural resources onto the planet and future generations.

However, to be able to slow down, quality of life beyond a certain base level must be decoupled from the increasing consumption of material goods and coupled with immaterial goods, perhaps related with more fundamental human values such as human relationships, trust, satisfaction at work, and a greater sense of shared meaning and purpose. Furthermore, as referred earlier, companies would have to stop externalizing their social and environmental costs and start accounting for the cost of using natural resources in the generation of value [13].

A number of social innovations, summarized in the following pages, appear to be fundamental in enabling a much-needed progressive slowdown.

First, one major social innovation would be to establish as human right the access to a universal basic income. Universal Basic Income—UBI—is a government-guaranteed payment that citizens receive. The intention behind the payment is to provide enough income to cover the basic living cost and provide financial security. While there are different approaches regarding who receives it (every citizen regardless of income versus only those who are below the poverty line, whether they are working or not), there are already a number of interesting experiments taking place in different parts of the world, such as Canada, Finland and Scotland. Opponents to UBI claim that it incentivizes idleness and a culture of dependency. However, existing experiences point out that the main benefits of UBI appear to involve the following key points [14,15]:

- Unemployed workers can afford to wait longer and search for a better job or better wages.
- Workers are disentangled from paid labor, and this helps to enhance creativity and social entrepreneurship.
- It gives adults the possibility to return to school or stay at home to care for their children and their relatives.
- It avoids the usual "poverty traps" created by traditional welfare programs.
- It allows for simple straightforward financial assistance that minimizes bureaucracy, an obstacle which often penalizes people with lower levels of education.
- The government would spend less administrating a universal income program than what it spends on traditional welfare assistance programs.
- Young couples have more income to start families, which is particularly relevant for countries with low birth rates.
- The payments could help stabilize the economy during recessionary periods.

Second, another much-needed social innovation is the adoption of new economic metrics. For example, there are few cases around the world where policymakers acknowledge that Gross Domestic Product (GDP) is a poor way to assess societal well-being. GDP measures mainly market transactions. It ignores public, social and environmental costs, and the cost of crime and income inequality, amongst other relevant indicators. Moreover, the use of GDP as the main indicator does not encourage developing countries to adopt more sustainable models of development [16]. Alternative measures of progress can be divided into three broad groups. The first group consists of adjusting economic measures for factors such as household work, income distribution, pollution and the depletion of natural capital. The second group of measures relies on perceptual indicators based on surveys of life satisfaction. The most comprehensive of these is the World Values Survey (WVS), which covers about 70 countries and includes questions about how satisfied people are with their lives. Finally, a third way to measure social–economic progress is to use weighted composite measures that combine indicators of health, income and living conditions with perceptual survey indicators. One example is the Happy Planet Index, introduced by the New Economics Foundation in 2006 [17]. While only a few countries adapted alternative measures of GDP, most notably Bhutan, there are interesting experimental approaches such as the Well-Being Budget recently introduced in New Zealand and the GEP—Gross Ecosystem Product—introduced in China, which measure the monetary value of ecosystem goods and services that benefit people—such as flood protection and clean water.

While surely none of these new composite measures of well-being are perfect, they offer the building blocks for something much better than GDP. However, creating a successor to GDP requires a sustained, transdisciplinary effort to integrate metrics and build consensus.

Finally, another direction for the development of social innovation, to complement the previous suggestions, would be any action capable of diminishing excessive consumerism, i.e., by helping decouple consumption from materialism. This can take at least two forms. One is to promote the sharing economy. Sharing economy envisions maintaining the level of products and services that can be consumed by multiplying the number of users per object. The central idea is to reduce material needs by optimizing their usage, and it is

characterized by the phrase: "You don't need a drill; you need a hole in the wall". By connecting people and businesses with the resources to those that want them, the sharing economy removes market inefficiencies, empowers consumers, and has the potential to impact positively a wide range of sectors.

However, we must go well beyond the sharing economy and enter into the promotion of conscious consumption. For instance, the prioritization of doing, seeing and feeling over possession is a fundamental change in consumers. Conscious consumers may, for example, trade with friends or shop for second-hand goods. Conscious consumers may look for fair trade certificates and/or certificates of "ethically sourced" and "social responsibility", such as the B Corps, or prefer to shop online using websites providing "time bank" solutions as an alternative currency system in which hours of service take the place of money.

Ultimately, conscious consumption is about new lifestyles for health and sustainability that value spiritual well-being over material products and possessions.

## 3. Finance and Banking as If People Mattered

According to the economist Bernard Lietaer [18], in 1975, roughly 80% of foreign exchange transactions involved the real trading of a product or a service. The remaining 20% were speculative transactions, i.e., bets made on the value of stocks or currencies going up or down—buy it before it rises, sell it before it drops. By 1997, the percentage of foreign exchange, which involved transactions in the real economy, changed dramatically to account for only 2.5%. In 2011, according to the Global Policy Forum, the picture was even starker, with only 0.6% of foreign exchange that could be traced back to a genuine international trade of goods and services. Of the rest, a minimum of 80% was directly attributable to speculation.

This disconnect between the financial and real economy produces financial bubbles that are at the root of global economic crises, e.g., the US real estate crisis in 2006, which was followed by the world financial crisis in 2008, and the Euro crisis. Money is a tool for the exchange of goods and services, and it cannot exist unless there are goods produced and resources to produce them; meaning, money has no value without the real economy that it relates to, or, to put it in other words, "Without Main Street, there is no Wall Street" [19].

However, the financial sector treats money as a product. Money itself has become the most traded and profitable product. While the profits of the financial sector have been growing over the last decades of the 20th century, does the financial sector really add much more value to the economy than it used to? Surely not. It just got used to considering "money" as a "product" and focusing on short-term financial profitability, thus ignoring the unintended side effects that damage the long-term health of the whole economic system. Furthermore, because financial institutions have become too large to fail, this situation has proven very difficult to change. With the complicity of central regulating banks, we have a financial system that privatizes profits but, when a crisis arises, socializes losses. Since 2007, more than 30 banks in Europe were bailed out, including Commerzbank (Germany), Fortis (The Netherlands and Belgium), KBC (Belgium), ABN e SNS Reaal (The Netherlands) and Royal Bank of Scotland (UK). The spectacular fall of one of the most prominent banks in Portugal (Banco Espírito Santo), which amounted for a total of 6.4 billion euros of accumulated debt, is also a good example of how a bank's losses need to be covered by public funding in order to avoid systemic problems in the financial system.

While today we have a more regulated system, the situation has not changed significantly since the bailout interventions from governments around the world following the 2008 financial crisis. We still have a system that accumulates an oversupply of money and capital in areas that produce high financial returns but low environmental and social returns. At the same time, we have a huge financial inclusion problem in third world countries and an undersupply of money and capital in areas that serve important societal and community investment needs, such as the education of children in low-income communities.

This creates opportunities for social innovation in finance and banking. There are a number of interesting cases of regenerative banking illustrating that finance and banking can be used not for short-term "extraction of value", but for supporting projects in all sectors of the economy. Banks such as Triodos Bank (perhaps the most famous sustainable bank) and BRAC Bank in Bangladesh serve as examples that regenerative banking can be profitable. Another example studied by the Capital Institute (a non-partisan, transdisciplinary and collaborative institute launched in 2010) is the First Green Bank in Florida.

## 4. Work with Meaning and Purpose

While unemployment was already one of the most perplexing problems in the economic system before COVID-19, today it is clear that the impact of the pandemic on jobs has been worse than expected—particularly in developing countries with no public means to support workers.

However, the problem is much more complex as the quality of work of those who managed to maintain their paid employment appeared to be downgrading even before the pandemic. In some sectors, value creation spreads over long and global value chains to the point that work contribution is so diluted that it loses "meaning". Human work needs meaning, and cannot be taken as "renting time" or "working for money", as people need to understand and connect (feel) their work (labor) contribution to the whole.

Today, there are clear signs that this dilution of work, together with a tremendous acceleration of life and work, often leads to multitasking, excess work and a permanent state of "fight or flight", thus causing exhaustion and loss of meaning. According to a recent report [20], the levels of physical, mental and spiritual "exhaustion burnout" (around 3.10 on a 1 to 5 scale in Europe) are increasing. This is the same for people suffering from CFS—Chronic Fatigue Syndrome—or depression and dependence on anti-depressive medications. CFS is different from burnout as it involves prolonged periods of physical and mental exhaustion of at least six months. Depression is also different from burnout and CFS as it tends to involve behaviors of lethargy and detachment.

The use of internet technologies, in particular for those working from home, appears to increase pressure as well as the emotional need to prove one's worth through one's job, thus leading to even more burnout and exhaustion. Synchronous and asynchronous communication tools, always buzzing with life, and the "available 24 × 7" culture are making it difficult for people to rest at any hour of the day or night. Without clearly defined time boundaries, many people tend to over-stretch themselves and increase the anxiety of underperformance—a sense of not being good enough or not being able to live up to expectations. Moreover, with so many online offers of new information and learning, FOMO—Fear Of Missing Out—is also becoming a serious problem for many people.

However, the obsession with work productivity is one of the major factors contributing to people moving away from long-term jobs to short-term contracts, temporary work and so-called "necessity" entrepreneurship. According to The Freelancing in America Survey, 35% of the American workforce, that is around 57 million people, were independent workers—freelancers, contractors or temporary employees—in 2019 [21]. Apparently, the rise of freelance work is not only because of a shortage of new jobs for younger people and corporate downsizing, but also the result of increasing employee dissatisfaction with traditional work lifestyles as most freelancers want to be freelancers in order to regain control of their time. The existence of online marketplaces for pairing peoples' talents with business needs is also a facilitator of this increase in independent freelance workers. Rather than engaging a person full-time with benefits and a salary, a company can find targeted and better-qualified talent to address their needs in this way, typically at lower costs. At the same time, for younger people, entrepreneurship appears to have a new meaning, leading to a higher number of younger freelance entrepreneurs in jobs that did not exist 10 years ago, such as digital business platforms managers, software programmers, graphic facilitators, digital video editors and social media curators.

With the COVID-19 pandemic eliminating many jobs or putting them on hold, there is further strain placed on some occupations such as those related to healthcare, elderly care, public transport and education, which are now considered "vital professions" despite being areas of public underfunding.

One important aspect that needs to be highlighted is that, in the future, there will be a probable shrinkage in the number of jobs. Even if the economy grows (and it cannot grow since we are already using 1.7 planets to provide the natural resources needed for the economy and absorb the waste), it would not produce the quantity and quality of jobs needed. If we count not just the unemployed but also the welfare-dependent, people at risk of poverty and social exclusion, the disabled and those reliant on other social benefits, many can be found to be in a difficult situation concerning income.

Following what we suggested in previous sections, if we assume that everyone has access to an income through work and/or through an UBI scheme, social innovation opportunities related to the quality of work would lie on how to find ways to connect work with meaning and purpose i.e., giving everyone a chance to find purpose in what they do and pursue their aspirations and dreams, thus putting their creativity in the service of a larger community.

Online trade platforms such as Worldstock (https://www.overstock.com/, accessed on 19 December 2021), which enables artisans from different parts of the world to display and sell their products, are a good example of how to avoid loss of meaning and purpose in long global value chains.

In order to help workers in general and freelancers in particular to find a better work–life balance, one opportunity would be to create new places to work, such as co-working spaces, creative hubs and quarters. The COVID-19 pandemic is already accelerating remote work (even for people working in large corporations) and it is likely that a large percentage of workers will not go back to their former work modality inside their employers' premises. However, to avoid feeling isolated, they may join a co-work in their residential area. Co-working spaces are a new type of social innovation that is not only capable of providing office space with good Wi-Fi but also creating a sense of community that comes from working around others. Moreover, in some cases co-working spaces offer a collaborative learning ambient that enhances individual and collective development, hence helping to create work purpose and meaning. As an example, we refer to the program "Exploring Our Town" supported by the National Endowment for the Arts in the USA [22].

## 5. Tech for Good

Creating social and economic value involves not just the use of capital and labor, as we saw in previous sections, but also the use of knowledge and technology. Nevertheless, while technology can and should be used as a tool to reduce social and ecological problems, it also brings new challenges.

Digital technology, and in particular the internet, is creating a new virtual "context" and reducing people as "context-responsive". For example, social media networks such as YouTube and Facebook filter out information that does not match each individual's worldview: they keep us in our echo chambers. Hence, people are at risk of being rapidly "decontextualized" to the point where they are so much in their virtual world that they do not notice the real and, in particular, lose the ability to "feel" what is real.

One way to capture this "decontextualization" is through the new definition of "digital divide". The digital divide used to be about access to the internet and social media, but now that a majority of the population in more developed countries have access to these technologies, the new digital divide is about limiting access to technology. For example, in education, the real digital divide is not between students who have access to the internet and those who do not, but rather between students whose parents know that they have to restrict screen time and those whose parents believe that more digitalization and more screen time is the key to success.

Apparently, the COVID-19 pandemic further contributes to this decontextualized society by pushing what Eric Schmidt (former vice president of Google) termed as the "no-touch society"—a society where COVID's physical distancing is reinforced by a "personalized digital search" [23] that keeps us addicted to our information bubble. The danger is that this virtual "screen society" is distracting from using localized community knowledge obtained from the engagement with local human interactions. As early as 1993, Howard Rheingold [24] was already calling attention to the ways online interactions are likely to affect human relationships and change our experience of the real world as individuals and communities. There are plenty of examples of how excessive exposure to "screens" and our "info-bubble" is causing a profound alienation nostalgia. For example, psychologists from the University of Wuerzrburg [25] argue that the use of smartphones to access social media networks while walking (i.e., being a smartphone zombie), which has become a prevalent phenomenon in many cities worldwide, is related to the Fear Of Missing Out (FOMO) and that smartphone use while walking, as a compensation for real human company, is sidelining the need to traverse safely.

In addition, with the COVID-19 pandemic, social media platforms like YouTube, Facebook, Instagram and TikTok, and communication tools such as Zoom and MSTeams, have moved from mediating the social fabric to becoming the social fabric. Social media and communication platforms have turned, in just a few months, into primary ways of interacting with others and making sense of the crisis, coordinating critical services, receiving support, and expressing their pains. Until now, these platforms hid behind the fiction of "neutrality"—that sorting the news feed by what people liked and shared would result in a richer information environment for all. However, not only is that fiction wrong, it is dangerous as well. Platforms such as Facebook, Instagram and TikTok are creating a race for attention. More than two billion people—a psychological footprint bigger than Christianity—are registered on social platforms designed with the goal of not just getting our attention, but also making us addicted to receiving attention from others. This creates an "extractive attention economy". In some cases, algorithms and games recommend increasingly extreme and outrageous topics and explore human weaknesses—fear, outrage, vanity, to keep us glued to the screen and exposed to online advertising.

Social media platforms also fuel a mass spread of disinformation, polarization and ultimate breakdown of truth—a breakdown that has made it harder to agree on obvious threats like COVID-19. The "fake news" phenomenon exemplifies the unforeseen consequences of this online social mediation by enabling the unethical mass subversion of public opinion faster than ever before.

One thing we can now see more clearly is that social media platforms are no longer used to serve the public interest. Instead of focusing their vast reach and existing capabilities to deliver useful information, support mental health, and enhance the capacity to find common ground and take collective action, they often amplify the collective sense of feeling helpless. To a large extent, these platforms helped create what Stephen Bertman [26] calls the "hyper culture" i.e., a chronic addiction to speed that is affecting value judgments and morals, often resulting in short-term choices that do not contribute to a sustainable future. They also enable what Shoshana Zuboff [27] calls "Big Other" or another form of "Surveillance Capitalism". "Big Other" is constituted by existing online mechanisms of extraction, commodification and control of information that effectively exile persons from their own natural behavior, while producing new markets of behavioral prediction and modification.

There is today another heavy trend, deeply entrenched in decades of science-push policy, which believes that science and technology can solve everything. This goes well beyond the effects of digitalization of society and economy, and includes other areas of knowledge such as biotech and nanotech. In particular, there is a general sense that the future lies in the use of science and technology to impose order on nature and society. One extreme example is that some people believe a giant space umbrella could help cool down the planet, i.e., that humans can modify the climate through space geoengineering [28].

Another example that technology is often taken as a solution for everything is the increasing smartification of products/services, and the production of useless gadgets [29]. Again, with the COVID-19 pandemic, the idea that technology can be used to enforce physical distancing or backtrack human interactions is contributing to the progress in "Surveillance Capitalism". One frightening example is how the police in Singapore are using robot dogs to patrol public spaces and keep people at a "safe" distance [30].

Furthermore, beyond "digitalization and "smartification", techno-optimism can also be seen in some form of "transhumanism" i.e., human faults and weaknesses able to be corrected by technology fixes. One example of this is Elon Musk's neural link, announced as capable of restoring eyesight, hearing and limb movement, together with its capability to address diseases that affect the brain [31].

The idea that science and technology can solve anything and will ultimately save human society is alarming, and may be related to the deep disconnect with nature and ourselves that we referred to earlier. Despite continuous efforts, technology may not save us, and rather than expanding individual and collective abilities, it may reduce wisdom and destroy many cognitive abilities. What we need is for technology to solve (real) social and human problems, instead of using it to disregard people's rights or take advantage of human weaknesses. Technology must approach innovation and design with an awareness of protecting and preventing the possible ways in which we are manipulated as human beings.

There are a few examples of initiatives on how to counteract the effects of internet technologies and social media. One that we would highlight is CHT—the Center for Humane Technology (https://www.humanetech.com/, accessed on 19 December 2021). CHT organizes open- and closed-door conventions for global leaders, promotes public testimonies to policymakers and heads of state, and manages mass media campaigns to help shift the mindset from which persuasive technology systems are built.

### 6. Transformative Learning and Education

In this section, we turn to the question of how social innovation can help advance and transform the current education system into a new system that truly supports a more ecological, equal and inclusive society.

In the last decades, the public education system was repackaged to conform to the philosophy of a quasi-market system. To do that, it adopted a utilitarian view and managerial practices taken from the New Public Management, or NPM, which translated into multiple certification schemes for conformity and in indexes and rankings of schools. As a result, current education is becoming mostly functional and based on information learning. While it recognizes a narrow part of human ability and focuses on understanding and memorizing figures, facts and formulas, it also fragments the understanding into knowledge domains such as health, environmental, economic and social sciences, to name a few. The main problem here is the disconnect between what the education system teaches, and the increasing complexity, interdependency and systemic perspective of societal and economic phenomena.

In contrast to a standardized certified view of education, we need a view that builds from humanistic values by taking advantage of new developments in learning theory and imperative of building a more equal and sustainable society. This corresponds to the willingness to let go of the instrumental view of education as "knowledge transmission" in favor of an education system that is centered on "transformative learning". Yet, this requires a clearer understanding of a relatively new view of the world concerning humanistic education. The realization of such a new education system requires a number of considerations. In the paragraphs below, we name a few of them for reference.

First, this new education system requires that the future of education be seen as learning-centric instead of teaching-centric or student-centric.

Second, it requires a change in context. Traditionally, the classroom, besides being defined as a physical space, has been a metaphor embodying assumptions concerning time,

length, pace, level and pedagogy. Nevertheless, learning also (and some will argue, mostly) happens outside the classroom, on the internet, and in the natural world. Therefore, there is a need to create a new context for learning and new metaphors. In other words, there a need to create new learning places outside the traditional classroom.

Third, not just the context needs to change but also the inner place of learning. According to Biester and Mehlmann [32], we need to educate people who can deal with complex social transformation processes and, in order to do that, they need to develop six key competences: self-knowledge, ability to work with people, envision, the capability of riding complexity, flow, and pedagogy. No collective change is possible without individual change, and yet, education systems do not work on individual change skills. It is particularly important to upgrade skills concerning attention, observation, listening, conversation and dialogue, and combine this with Science, Technology, Engineering and Mathematics (STEM) skills, in order to be able to deal with the current dynamic complexities of the world. On the other hand, we also need to progress from "big data" to "deep data", i.e., learn how to go beyond cause–effect association and recognize as well as interpret patterns so that we can enhance our ability to deal with disruptive transformation. In current society, we need people (and particular leaders, as we will explore in the next section) with skills related to seeing whole systems (see and sense!) and not just partial views of related variables.

Fourth, and to a substantial extent, the most important skill today appears to be "learning to learn". According to Lisa Christensen, Jake Gittleson and Matt Smith [33], learning itself has become a fundamental skill. Perhaps heutagogy or self-determined learning (where students determine their own learning trajectory), frequently referred to in the domain of e-learning, should span to other areas as a way to stimulate learning to learn [34]. In addition, there is today a greater need to promote reflexive learning (double-loop learning). This deep learning is much needed to enable the student to see things differently and promote an evolution of consciousness at the individual and group level [35]. It involves creativity and deep awareness of alternative worldviews. As Einstein suggested, it requires a shift in consciousness leading to a transformative learning experience that questions society's core values.

Fifth, it requires explicit focus on systemic thinking rather than linear thinking and with integrative learning rather than fragmentary learning by discipline. As pointed out by the Brundtland report [36] a long time ago, there are no separate problems of health, environment and economy; they are all interrelated. Hence, there is a need to integrate areas and stop fragmentation in the way we learn. Furthermore, we need to be more concerned with process dynamics rather than linear cause–effect equations. In other words, learning needs to include understanding aggregate complex patterns rather than minor associations between variables.

Sixth, it requires seeing school, curricula and community as completely interrelated. Moreover, the whole system approach to learning needs the integration of rational intelligence with human compassion and empathy. It also requires the integration of individual and collective creativity. It demands skills in deep listening, dialogic processes, mindfulness, social presence, as well as learning how to connect people around projects and initiatives.

The new system cannot, however, be mistaken for some simplistic approach that essentially is an extension of the current system and does not address the need for deeper and broader learning. For example, the inclusion of ethical issues in bioeconomy and sustainability finance, or in economics and management teaching, may be an encouraging start, but because socioeconomic thinking remains associated with mainstream linear thinking, these initiatives are missing the opportunity to use education as leverage for social innovation and as an instrument for transformation. According to Stephen Sterling [37], education serves four functions:

1. To replicate society and culture and promote citizenship—the socialization function.
2. To train people for employment—the vocational function.
3. To develop the individual and his/her potential—the liberal function.

4.    To encourage change towards a fairer society and better world—the transformative function.

The first two functions tend to stress an instrumental vision fueled by the current technocratic managerialist view of education. But changing the education system requires reinforcing intrinsic values and not just instrumental values. It needs a view of education not just as a means to achieve an end but as an end in itself. Changing education would therefore require a new balance in the four roles or functions, with the liberal and the transformative functions needing to be reinforced.

There are different examples and opportunities for social innovation involving education at different levels. In early childhood education, the Reggio Emilia Philosophy coattails off the innate curiosity of children and aims to assist them through the understanding of their world and who they are in it [38]. Children are able to pursue their own interests and build upon ideas at their own pace. Alternative models of higher education, such as the famous Alternative University in Bucarest and others, are also interesting experiments.

## 7. Leadership, Governance and the Need for New Coordination Mechanisms

Leadership and new governance mechanisms are nowadays key aspects in the transition to a more ecological and inclusive society. In fact, unforeseen disruptive events are likely to be more frequent in today's world, and therefore, leaders in general and high-level leaders in particular face new challenges but often appear to fail to anticipate change. The increasing frequency of leadership failure suggests that the established ways of decision making in major companies, civil servants or ministerial cabinets are no longer working. To illustrate this concept, let's look at a few examples.

The financial and economic crisis in 2008, the precipitous 60% fall in oil prices in early 2014, and the rise of the Islamic State with the capacity to seize Mosul in Northern Iraq are all clear examples of leadership failure. Currently, we have a huge refugee crisis at the EU borders, causing massive humanitarian problems, and yet European leaders are failing to find an effective and coordinated response. COVID-19 started an unprecedented global health crisis, but world leaders and organizations such as the WHO rejected early signs that the infection in China was going to lead to a global pandemic.

In the private sector, there are many examples of leadership failure too. In their report "Thinking the Unthinkable", Nick Growing and Chris Langdon [39] argue that today's corporate leaders agree that their mindsets, behaviors and systems are rarely adequate to handle an increasingly complex world. Furthermore, the leaders they interviewed claimed that their very organizations created barriers for them to hide behind, thus helping them defend their position rather than welcome and understand disruptive changes and the need to adapt and evolve their companies.

But why do leaders (both in the private and public sectors), despite the use of multiple scenario planning and foresight approaches, fail to anticipate and read the possible future? Leadership failure results from a close mindset rooted in the inability to see the reality. This is further aggravated by leaders' inability to feel what others feel (empathize), i.e., not just understand the system but also not sense and "feel" the system. As argued by Hans Rosling and others [40], the inability to see and sense is related to a tendency to worry about everything all the time, instead of embracing a worldview based on facts and on the correct sensing in relation to those facts. Data evidence and signals are often available, and yet they are ignored, as if the leaders and those who work for them are in constant denial. Following Rosling's argument, we need not have just an adequate focus on "factfulness" but also "mindfulness". The deficit of insights probably results from our inability to see problems from a different perspective.

Failure to anticipate change is further damaged by social media mechanisms (as referred to in previous sections) that amplify digital eco-chambers. For example, events such as the Notre Dame fire in April 2019 attracted widespread media attention but the heavy rain and strong winds caused by two cyclones, Idai and Kenneth, through March

and April 2019 in Mozambique, which led to flooding, hundreds of deaths and massive destruction of property and crops, did not attract the same attention.

We believe that leading business and economic transformation towards more human-centered capitalism, towards a socioeconomic system where human and social capital play a central role, requires leaders who are able to broaden their perspective on the whole social and economic system. Opportunities for social innovation may therefore relate to the need to design and implement new approaches towards leadership training focused on personal development, thus leading oneself towards the improvement of relations with others and how the leader relates to the whole system. The case of the ULab 1x online training from the Presencing Institute at MIT is an interesting example of a new approach to leadership based on a framework known as "Theory U"—a process that enhances human capacity to "presence" and to "pre-sense" an emerging future [41].

Finally, and in relation to the ability to see the whole system (from the edges of the system), a humanized and participatory form of leadership needs to be supported, not just by the traditional multi-stakeholders' consultation processes, and involve dialogic practices [42]. This is particularly important for the corporate world. Social innovation for large corporations is often taken as a set of practices related to "corporate social responsibility". However, social innovation in corporations needs to be related to fundamental changes in how the decision process is organized. For example, if corporations want to contribute to the 17 SDGs (Sustainable Development Goals defined by the United Nations in 2015), they cannot have the same hierarchical structures. They must progressively strengthen dialogic practices and adopt a new perspective of governance that goes beyond stakeholders' well-being and truly embrace society's well-being. Dialogic practices may, however, be facilitated by the use of "open space technology" [43] and practices such as Art of Hosting—AoH, World Café [44]. Other frameworks such as Appreciative Inquiry [45] or even Theory U, referred earlier, may also be useful.

## 8. A Final Comment: Disruptive Social Innovation and Opportunities for Social–Economic Transformation

This article, based on a non-systematic literature review around six key dimensions, displays a number of important limitations. First, our review is not comprehensive and therefore has selection bias. Second, it may also lack appropriate critical appraisal of the included studies. Finally, we also hope that our efforts at synthesis do not introduce further bias.

Nevertheless, the review article reflects a number of problems that we believe create new opportunities for disruptive social innovation—innovation that truly triggers wider processes of social and economic transformation. We looked first at problems related to the current economic model. The current economic-growth paradigm feeds on the non-accounted value extracted from nature—natural capital—and largely reflects the increasing disconnect between humans and nature. Second, another great source of economic and social problems is the disconnect between finance and the real economy. The undersupply of capital to finance the "commons" in areas of fundamental societal and community needs creates opportunities for social disruptive innovation. Third, the current socioeconomic system is downgrading human work. There is a need to redefine work-lifestyles so that they contribute to meaning and purpose. Fourth, another major source of opportunities for social innovation is how we use technology. The harmful use of technology is contributing to a "decontextualized" society, and to the so-called "extractive attention economy". Fifth, we also looked at education as a domain where social innovation is most needed. There is a very poor fit between what the current education system teaches, and the increasingly complex, interdependent and interdisciplinary perspectives needed to understand the current multiple social and economic transitions. This creates new opportunities for disruptive innovation in education. Finally, we also looked at how a deficit of leadership contributes to reinforcing current problems and demands for new leaders and new leadership training.

In many cases, innovative solutions to these issues are being attempted. While existing institutions are perhaps slower to respond, in some cases, both individuals and (private, public, philanthropic, and non-profit) organizations, dissatisfied with the above dysfunctions, are putting aside self-interest and coming together to challenge conventional approaches and propose innovative solutions to major world problems. Some believe that these spontaneous reactions correspond to a sort of Earth immunity system responding to worldwide challenges [46].

Still, we need to reinforce these initiatives. There is a need to amplify social innovation activism around economic human rights (such as basic income, access to health, education and entrepreneurial opportunity) in order to enable all people to carry out their full potential for entrepreneurial creativity, which generates wealth and social well-being. In an interesting study on how creative industries like fashion, art and music drive the economy of New York, Elizabeth Currid [47] argues that such activities drive the economy just as much, if not more, of other sectors such as finance, real estate and law. The so-called "Warhol Economy" is fueled by the social life that whirls around a certain cultural underground, where creative people meet, network, exchange ideas, pass judgments and set the trends that shape popular culture.

The use of technology and, especially, the accelerating digitalization of work need to be mediated by a human-values-based approach—a neo-humanistic approach. For example, in a post-COVID world where working online is likely to grow significantly, there will be a need to change the patterns of working and living. Different kinds of "work–life spaces", or common "work ecosystems" and co-work spaces, are therefore likely to be one of the most important social innovations for the next decades. Another example is that of technology in education, and how it should not be used to extend the current system but, instead, be used as an enabler of learning environments in ways chosen by learners, teachers, or co-created jointly. These new learning contexts are not about technology per se. They are about the use of new e-enabled techniques designed for new learning methods.

Pandemics have been important drivers in human history, and a catalyst for change [48]. The COVID-19 pandemic may increase social innovations as it affects the economy not only with a deep supply shock but also by the acceleration of technological development and adoption [49]. This is a decisive moment for social innovation both in research and practice [50] as a means for economic reforms to deal with emergent social needs or, in a more profound way, stimulate a deeper systemic change. A broader comprehension of transformation is necessarily associated with territorial social responsibility linked to positive choices of ecological and social innovation.

We are not all in the same boat as was initially argued by many [51]. COVID-19 skillfully took advantage of inequalities, exclusions and vulnerabilities. The pandemic marks the beginning of the 21st century as a cruel pedagogue by revealing in a dramatic way that contemporary society—based on principles of domination such as capitalism, colonialism and patriarchy, and convinced that nature belongs and is at the service of humanity—is largely inconsequential. [52]. Capitalism has been quick to profit from the pandemic by exacerbating inequality, but can the pandemic be the foundation of a paradigm shift? What would take a long time to occur is being precipitated by the pandemic. None of the current problems, from ecological, financial, poverty or unemployment, can be solved in isolation. While effective social innovation calls for integrated approaches, we also argue that profound collective change can only be achieved through individual change, i.e., through deep processes of personal transformation that will allow social innovation opportunities to emerge from this need to help individuals with their inner-self transitions. Society cannot waste yet another opportunity generated by a crisis [53]. But do not take any change for granted. Progress is reversible [54]. We cannot go back to normal because normality was the problem [55]. The future remains open.

**Author Contributions:** Conceptualization, M.L. and H.P.; methodology, M.L.; validation, H.P.; investigation, M.L. and H.P.; resources, H.P.; writing—original draft preparation, M.L.; writing—review

and editing, M.L. and H.P.; visualization, H.P.; project administration, H.P.; funding acquisition, H.P. All authors have read and agreed to the published version of the manuscript.

**Funding:** This research was funded by the Atlantic Social Lab—Atlantic Cooperation for the Promotion of Social Innovation, co-financed by the European Regional Development Fund (EAPA_246/2016) through the INTERREG Atlantic Area Cooperation Programme. Hugo Pinto acknowledges the support of the Portuguese Foundation for Science and Technology (FCT) through the Scientific Employment Support Program (DL57/2016/CP1341/CT0013).

**Institutional Review Board Statement:** The study was conducted according to the guidelines of the Declaration of Helsinki.

**Acknowledgments:** This review article is an improved and updated version of a report of a project report prepared in the context of the Atlantic Social Innovation Observatory (http://atlanticsociallab. eu/, accessed on 19 December 2021), Work Package 6 of the Atlantic Social Lab project.

**Conflicts of Interest:** The authors declare no conflict of interest.

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
