# Peer review of "Transformation for a Post-Pandemic World: Exploring Social Innovations in Six Domains"

_knowledge, doi:10.3390/knowledge2010010_

Round 1
Reviewer 1 Report
The paper is generally well written and structured. Based on their analysis, I suggest that authors discuss some major research streams in the last section of the manuscript. I also suggest to cite more relevant and recent literature to support reasoning.
Author Response
The paper is generally well written and structured.
Authors: Thank you.
Based on their analysis, I suggest that authors discuss some major research streams in the last section of the manuscript.
Authors: we have inserted some text in the conclusion to respond to this request
I also suggest to cite more relevant and recent literature to support reasoning.
Authors: we have added some references through-out the paper.
Reviewer 2 Report
the paper analyses an interesting and contemporary issue, but it needs to be improved.
I particularly recommend:
- the abstract could be supplemented by highlighting more the methodology used and the peculiarities of the paper. in addition to the key words I would also add well-being
- as far as the introduction is concerned, I think it is essential to specify the basis for the breakdown of social innovation in figure 1.the section on explaining the structure of the paper is well written and clear.
- The conclusions need to highlight the limits of the research and, more importantly, future prospects. I would also link these to the concept of social and ecological innovation and the territorial social responsibility linked to positive choices of social innovation.
- there is a need to further integrate the literature on social innovation related articles during the covid 19 period.
- I recommend such papers to be added to the literature
Cattivelli, V., & Rusciano, V. (2020). Social innovation and food provisioning during COVID-19: The case of urban–rural initiatives in the province of Naples. Sustainability, 12(11), 4444.
Rusciano, V., Civero, G., & Scarpato, D. (2020). Social and ecological high influential factors in community gardens innovation: An empirical survey in Italy. Sustainability, 12(11), 4651.
Rusciano, V., Scarpato, D., & Civero, G. (2019). TERRITORIALSOCIAL RESPONSIBILITY: A CLUSTER ANALYSIS ON A CASE STUDY. Calitatea, 20(S2), 543-548.
Author Response
The paper analyses an interesting and contemporary issue, but it needs to be improved.
Authors: Thank you.
I particularly recommend:
- the abstract could be supplemented by highlighting more the methodology used and the peculiarities of the paper. in addition to the key words I would also add well-being
Authors: abstract was revised and keyword added.
- as far as the introduction is concerned, I think it is essential to specify the basis for the breakdown of social innovation in figure 1.
- the section on explaining the structure of the paper is well written and clear.
- The conclusions need to highlight the limits of the research and, more importantly, future prospects. I would also link these to the concept of social and ecological innovation and the territorial social responsibility linked to positive choices of social innovation.
- there is a need to further integrate the literature on social innovation related articles during the covid 19 period.
- I recommend such papers to be added to the literature
Cattivelli, V., & Rusciano, V. (2020). Social innovation and food provisioning during COVID-19: The case of urban–rural initiatives in the province of Naples. Sustainability, 12(11), 4444.
Rusciano, V., Civero, G., & Scarpato, D. (2020). Social and ecological high influential factors in community gardens innovation: An empirical survey in Italy. Sustainability, 12(11), 4651.
Rusciano, V., Scarpato, D., & Civero, G. (2019). TERRITORIALSOCIAL RESPONSIBILITY: A CLUSTER ANALYSIS ON A CASE STUDY. Calitatea, 20(S2), 543-548.
Authors: we have tried to include these suggestions. We also have analysed these references and included when we found them pertinent for our narrative.
Reviewer 3 Report
The work is an attempt to structure the tide of recent times for the readers, however, lacks the quality of either a scientific paper or adding any substantial value to existing knowledge in the field. Several sections are hard to read as it is hard to tell which bits are thoughts of the author/s and which ideas are contributions of other peers in the field. Missing references is a huge issue throughout the work. Figure 1 could have been more sound and integrated if the work was built on the works of other established frameworks in the field. At the moment, the piece reads more like an essay than a research paper to fit a journal of this stature. Some information is common knowledge and they do not necessarily contribute to the 'frame' that the paper tries to provide. It would be a lot better if existing frameworks are reviewed adequately, relevant data is collected to validate the claims made in the paper, along with a more concise explanation of the elements of figure 1 and more importantly how these elements are entwined.
Author Response
The work is an attempt to structure the tide of recent times for the readers, however, lacks the quality of either a scientific paper or adding any substantial value to existing knowledge in the field. Several sections are hard to read as it is hard to tell which bits are thoughts of the author/s and which ideas are contributions of other peers in the field. Missing references is a huge issue throughout the work. Figure 1 could have been more sound and integrated if the work was built on the works of other established frameworks in the field. At the moment, the piece reads more like an essay than a research paper to fit a journal of this stature. S
Authors: This text is a review, much closer to an essay than a regular research article. Despite this fact, we still believe this contribution is relevant to stimulate the ongoing debate about the post-pandemic transformation. More references were included. Currently the text includes 49 references, many of them recent.
Some information is common knowledge and they do not necessarily contribute to the 'frame' that the paper tries to provide.
Authors: Thank you. We believe that some of this common knowledge ideas are crucial to create a base of common understanding with the reader in order to create a better narrative in each sub-section.
It would be a lot better if existing frameworks are reviewed adequately, relevant data is collected to validate the claims made in the paper, along with a more concise explanation of the elements of figure 1 and more importantly how these elements are entwined.
Authors: We agree with the reviewer that these suggestions are relevant, but we feel that this implies another paper, much more empirical, not necessarily this one, that is an essay. Nonetheless, we tried to include some sentences to underline these limitations in the text.
Round 2
Reviewer 1 Report
The authors have satisfactorily addressed all my concerns. Their paper can be published.
Reviewer 2 Report
the paper is improved and suitable for publication.